# Seasonal Patterns of Picocyanobacterial Community Structure in the Kuroshio Current

**DOI:** 10.3390/biology12111424

**Published:** 2023-11-13

**Authors:** Ya-Fan Chan, Chih-Ching Chung, Gwo-Ching Gong, I-Jung Lin, Ching-Wei Hsu

**Affiliations:** 1Department of Microbiology, Soochow University, Taipei 11101, Taiwan; yfchan.micro@gmail.com; 2Institute of Marine Environment and Ecology, National Taiwan Ocean University, 2 Pei-Ning Road, Keelung 20224, Taiwan; gcgong@mail.ntou.edu.tw (G.-C.G.); linijung65@gmail.com (I.-J.L.); mamies14@gmail.com (C.-W.H.); 3Center of Excellence for the Oceans, National Taiwan Ocean University, 2 Pei-Ning Road, Keelung 20224, Taiwan

**Keywords:** prokaryotic picophytoplankton, Kuroshio current, *Synechococcus*, *Prochlorococcus*

## Abstract

**Simple Summary:**

The cell enumeration, 16S rRNA phylogenetic analysis, and hydrography determination were conducted to reveal the ecology of prokaryotic picoplankton in the subtropical Kuroshio current. The picocyanobacteria (i.e., *Synechococcus* and *Prochlorococcus*), contributing more than 50% of chlorophyll *a*, were important primary producers in the subtropical Kuroshio current. The notable seasonal distributions of picocyanobacteria and hydrography were also well described. We suggested the ambient nutrient contents should be the crucial parameter to determine the seasonal patterns of *Synechococccus* and *Prochlorococcus* in the study area. Because of the ability to compete for nutrients in an oligotrophic environment, picocyanobacteria would become dominant primary producers in marine ecosystems under the scenario of global warming.

**Abstract:**

The nutrient-scarce, warm, and high-salinity Kuroshio current has a profound impact on both the marine ecology of the northwestern Pacific Ocean and the global climate. This study aims to reveal the seasonal dynamics of picoplankton in the subtropical Kuroshio current. Our results showed that one of the picocyanobacteria, *Synechococcus*, mainly distributed in the surface water layer regardless of seasonal changes, and the cell abundance ranged from 10^4^ to 10^5^ cells mL^−1^. In contrast, the maximum concentration of the other picocyanobacteria, *Prochlorococcus*, was maintained at more than 10^5^ cells mL^−1^ throughout the year. In the summer and the autumn, *Prochlorococcus* were mainly concentrated at the water layer near the bottom of the euphotic zone. They were evenly distributed in the euphotic zone in the spring and winter. The stirring effect caused by the monsoon determined their distribution in the water column. In addition, the results of 16S rRNA gene diversity analysis showed that the seasonal changes in the relative abundance of *Synechococcus* and *Prochlorococcus* in the surface water of each station accounted for 20 to 40% of the total reads. The clade II of *Synechococcus* and the High-light II of *Prochlorococcus* were the dominant strains in the waters all year round. Regarding other picoplankton, *Proteobacteria* and *Actinobacteria* occupied 45% and 10% of the total picoplankton in the four seasons. These data should be helpful for elucidating the impacts of global climate changes on marine ecology and biogeochemical cycles in the Western Boundary Currents in the future.

## 1. Introduction

Picophytoplankton, including picocyanobacteria and picoeukaryotes, are the phytoplankton whose cell size is smaller than 2 μm. Because of their high surface area to volume ratio, picophytoplankton prevail in diverse marine environments, especially thriving in oligotrophic oceans. Generally, marine phytoplankton contribute approximately half of global primary production, which is about 46 to 50 petagrams per year. Of these, picophytoplankton are responsible for approximately 24% of marine phytoplankton production [1,2]. In particular, picophytoplankton are the major primary producers in some marine provinces [3]. For example, picophytoplankton contribute to greater than 70% of the total primary production in the tropical Pacific Ocean [4]. Furthermore, picophytoplankton in the Mediterranean provided 31% to 92% of the primary productivity [5]. The biomass of picophytoplankton is rapidly consumed by heterotrophic microorganisms and enters the grazing food chain or the microbial loop. Several studies have predicted that ocean warming may cause the fraction of tiny phytoplankton (picophytoplankton) to increase over that of the larger group (>2 μm) [6,7]. This relatively rapid change might alter the phytoplankton communities toward picophytoplankton sizes. Then, it may further change the functioning and biogeochemistry of pelagic ecosystems. Therefore, picophytoplankton are important primary producers in marine ecosystems and global oceanic biogeochemistry cycles [3,8].

*Synechococcus* and *Prochlorococcus* are the significant members of picocyanobacteria. The abundances of *Synechococcus* and *Prochlorococcus* in the global oceans are approximately 10^4^ to 10^6^ cells mL^−1^ [9,10,11,12]. However, due to their different biological characteristics, *Synechococcus* and *Prochlorococcus* have been suggested to occupy diverse marine provinces. *Synechococcus* is distributed in the global oceans [9,13]. Based on genetic information, *Synechococcus* has been divided into fifteen clades and twenty-eight subclades [14]. Clades I and IV were found in cold and nutrient-rich waters, whereas *Synechococcus* clades II, III, V, VI, and VII frequently appear in tropical and subtropical waters [12]. *Prochlorococcus* usually distributed the euphotic zone of tropical and subtropical oligotrophic waters between latitudes 40° N and 40° S [15,16]. Based on their distribution in the water column, *Prochlorococcus* species are categorized into two groups: high-light (HL)- and low-light (LL)-adapted ecotypes [14]. These two ecotypes present distinct distributions across depths [17,18,19]. HL-adapted ecotypes have a lower divinyl chlorophyll b/divinyl chlorophyll *a* (Chl *b*/Chl *a*) ratio and are typically found in the surface waters of the open ocean [20]. LL-adapted ecotypes have a higher Chl *b*/Chl *a* ratio and usually grow in deep waters with much lower light intensities. In addition to the different distributions in the water column, the appearance of two *Prochlorococcus* ecotypes also exhibits distinct seasonal dynamics. For example, the abundance of LL-adapted ecotypes has no apparent seasonal variation in the northern Red Sea, in contrast to the distribution of HL-adapted ecotypes [21]. Moreover, the HL-adapted ecotypes are composed of six clades. HL clade II was found to dominate at low and midlatitudes but was changed by HL clade I at latitudes above 30° [16,22]. Furthermore, it has been demonstrated that sudden extreme climate events, such as Asian dust and typhoons, temporarily change the picophytoplankton community composition [23,24].

The Kuroshio current is one of the western boundary currents located in the Northwest Pacific Ocean. The westward-flowing North Equatorial Current runs into the Philippine coast and then bifurcates into northward (i.e., the Kuroshio current) and southward streams (i.e., the Mindanao Current). The Kuroshio current passes the east coast of Taiwan and southeast coast of Japan and then continues as the North Pacific Current [25,26]. The width of the Kuroshio mainstream in eastern Taiwan is approximately 100 km, the depth is around 800 to 1000 m, and the mainstream speed is approximately 1 to 1.5 m s^−1^. The Kuroshio water is high-temperature and high-salinity. In the subtropical segment of the Kuroshio, its temperature and salinity have no significant seasonal variation and remain at 26 °C to 30 °C and approximately 34.5, respectively [27]. The Kuroshio water in the euphotic zone is characterized as ultraoligotrophic [28,29]. However, when the Kuroshio current flows near the east coast of Taiwan, upwelling along the coastal region transports deep nutrients into the surface [29,30]. The uplift of deep seawater provides a large amount of nutrients to the Kuroshio euphotic zone when the Kuroshio invades the East China Sea shelf [31]. In addition, terrestrial material injection and vertical mixing caused by typhoons or the northeast monsoon provide nutrients to the Kuroshio current. These nutrient sources support primary productivity in the Kuroshio region [32,33]. The surface Kuroshio is oligotrophic and has low chlorophyll *a* (Chl *a*) concentrations of less than 0.2 mg m^−3^ [29,34,35,36,37]. The yearly primary production in the subtropical Kuroshio current is approximately 0.91 ± 0.47 mg carbon m^−3^ day^−1^ [36]. Several studies have presented a significantly positive correlation between Chl *a* and the abundance of picophytoplankton in the Kuroshio [37,38]. Furthermore, various models of global circulation have suggested that the abundance of picophytoplankton will increase significantly with the rise of seawater temperature [39,40]. According to the above information, we hypothesize that with the ecology of picophytoplankton and the biogeochemical cycles, their lead should change in the subtropical Kuroshio under the scenarios of global warming. However, the assemblage composition of picophytoplankton is very complicated, and the relationship between their community succession and the primary productivity in this area requires further careful exploration. The seasonal dynamics of picophytoplankton in the subtropical Kuroshio current were completely revealed in this study. Our results will facilitate future studies on the picophytoplankton ecology and their relationship with the marine biogeochemical cycles under global climate change.

## 2. Materials and Methods

### 2.1. Sampling Scheme

This study is a long-term observation of phytoplankton community composition in the subtropical Kuroshio Sea from 2009 to 2015. Intensive sampling in four seasons was conducted from 2012 to 2013. A total of nine stations across the Kuroshio current were visited onboard the research vessels *Ocean Researcher I* and *Ocean Researcher II* in October (defined as the autumn voyage) 2012, January (defined as the winter voyage) 2013, April (defined as the spring voyage) 2013, and July (defined as the summer voyage) 2013 (Figure 1). Temperature and salinity were continually recorded by a conductivity/temperature/depth recorder (CTD) (SBE9/11 plus) (Sea-Bird Electronics, Bellevue, WA, USA). The current velocity was measured by a 150 kHz shipboard acoustic Doppler current profiler (ADCP; Teledyne RD Instruments, Poway, CA, USA) [41]. To determine the concentrations of chlorophyll *a* (Chl *a*) and inorganic nutrients and the abundance of picophytoplankton, water samples were collected using 20-L GO-Flo bottles (General Oceanics, Miami, FL, USA) mounted on the CTD rosette sampler at six depths from 5 m to 300 m. Light intensity in water was continually measured by a photosynthetically active radiation irradiance sensor (PAR sensor) (Chelsea Technologies, Molesey, UK) equipped with a CTD. The water samples collected from surface (depth = 5 m) and the deep chlorophyll maxima (DCM), which was defined by the fluorescence profile determined by the fluorometer (AquaTracka III, Chelsea, London, UK), at Stations K2, K4, K5, K6, and K8 were used to isolate environmental DNA for analyzing the phylogenetic diversity of the 16S ribosomal RNA (rRNA) gene.

### 2.2. Determination of Chl a and Inorganic Nutrient Concentrations

The detailed methods for determining the concentrations of Chl *a* and inorganic nutrients are described in Chan et al. [42]. Briefly, one liter of water for Chl *a* analysis was immediately filtered through a GF/F class filter (Whatman, Maidstone, UK) and stored at −20 °C until analysis. The Chl *a* retained on the filter was extracted in 90% acetone. The Chl *a* concentration was determined by a fluorometer (10-AU-005) (Turner Design, Charlotte, NC, USA)) [42]. To evaluate the fraction of Chl *a* contributed by picocyanobacteria to the total Chl *a* concentration, we applied 1.258 femtograms (fg) Chl *a* cell^−1^ and 1.203 fg Chl *a* cell^−1^ as the conversion parameters for *Prochlorococcus* and *Synechococcus*, respectively, which were determined by the average values of the Chl *a* concentration of several algal strains measured by high-performance liquid chromatography (HPLC) [43].

To determine the concentrations of nitrate (NO_3_) and phosphate (PO_4_), the water sample (100 mL) was placed in a polypropylene bottle, immediately frozen with liquid nitrogen and stored at −20 °C until analysis. The NO_3_ and PO_4_ concentrations were measured by the pink azo dye and the molybdenum blue methods, respectively. The detection limits of NO_3_ and PO_4_ are 0.3 and 0.01 μM, respectively [44,45,46]. The nitracline depth was defined as the depth at which the NO_3_ concentration difference was 0.5 µM concerning the surface value [47,48,49].

### 2.3. Determination of Picophytoplankton Abundance

The cells were fixed with paraformaldehyde at a final concentration of 0.2% (*w/v*) and were preserved in liquid nitrogen. Different picophytoplankton populations were categorized with flow cytometry (FACSAria) (Becton-Dickinson, Franklin Lakes, NJ, USA) based on cell size and autofluorescence in the range of orange from phycoerythrin (575 ± 15 nm, for determining *Synechococcus*) and red from chlorophyll (>670 nm, for determining *Prochlorococcus* and photosynthetic picoeukaryotes) under excitation at 488 nm. A known number of fluorescent beads (TruCOUNT tube) (Becton-Dickinson, USA) were parallelly calculated to convert the original cell abundance in the sample [24]. The putative relationships between the picophytoplankton distribution and ambient hydrographic characteristics were analyzed by redundancy analysis (RDA) and the envfit function in the vegan package in R.

### 2.4. Picoplanktonic DNA Isolation

Seawater was filtered through a 5-µm mesh nylon net to remove larger plankton. The planktonic cells in the filtrate were collected by 0.2-µm pore size polycarbonate membranes (Nucleopore) (Whatman, Stockbridge, GA, USA) under gentle vacuum (≤100 mmHg). The membranes were immediately frozen in liquid nitrogen until DNA isolation. The cells retained on the membranes were disrupted with lysozyme (Roche, Basel, Switzerland) and proteinase K (Roche, Switzerland) treatments. After purification by hexadecyltrimethylammonium bromide and phenol/chloroform/isoamyl alcohol (25/24/1, *v/v/v*), the DNA pellet was precipitated using isopropanol and resuspended in Tris-EDTA buffer (pH 8.0) [24]. The DNA concentration and purity were determined by spectrophotometry (NanoDrop) (Thermo Scientific, Waltham, MA, USA)) at wavelengths of 230, 260, 280, and 320 nm.

### 2.5. The Diversity of Picoplankton Composition

DNA product (10 ng) was used as the template for the polymerase chain reaction (PCR) to specifically amplify the V3–V4 region fragments of 16S rRNA genes using the high-fidelity DNA polymerase 2× KAPA HiFi HotStart ReadyMix (Roche) and the forward primer 16SV3V4-F (5′-tcgtcggcagcgtcagatgtgtataagagac AGCCTACGGGNGGCWGCAG-3′) and the reverse primer 16SV3V4-R (5′-gtctcgtgggctcggagatgtgtataagagacAGGACTACHVGGGTATCTAATCC-3′) (the lowercase letters indicate the Illumina adaptor sequences; W=A or T; H=A, T or C; V=A, C or G; N=A, T, G or G) [50]. The amplicons were analyzed by the Illumina MiSeq high-throughput nucleotide sequencing platform (Illumina, San Diego, CA, USA)) using the pair-end method. After the removal of low-quality reads (quality score > Q20) and the trimming of adaptor and primer sequences by Cutadapt software 4.6 [51], the diversity of the resultant reads was analyzed by DADA2 software 1.26 [52]. The taxonomic assignment of representative amplicon sequence variants (ASVs) obtained by DADA2 analysis was further conducted with the 16S rRNA reference sequences in the Silva database (version 138). The ASVs of *Synechococcus* and *Prochlorococcus* were assigned following the previous study [53].

The indices of richness (abundance-based coverage estimator, ACE) and diversity (Shannon) were estimated by the sequences randomly subsampled (the size of the smallest library was used) from each sample 1000 times and were expressed as averages to avoid biases generated by differences in the sequencing depth. The beta diversity analysis was performed with hierarchical clustering to visualize the spatiotemporal distribution of the picoplankton community assemblage by PRIMER 6 software with the Bray–Curtis distance.

### 2.6. Nucleotide Sequence Deposition

The nucleotide sequences used in this study have been deposited in the Sequence Read Archive (SRA) database under BioProject accession number PRJNA904529.

## 3. Results

Based on physical characteristics such as current velocity, temperature, and salinity, stations were divided into four categories, namely: K1, which was the station with coastal upwelling; Stations K2 to K4, which were located at the mainstream of the Kuroshio current; and Stations K5 and K6, which were close to the Kuroshio, and their hydrological characteristics were affected by it. Stations K7 to K9 were located in the open ocean and were regarded as the reference, where they were not affected by the Kuroshio current [42,46]. The hydrography in the upper water column (≤100 m) of the Kuroshio is high-temperature, high-salinity, and low-nutrient content [46]. In addition to Station K1 being primarily affected by upwelling, the intensity of monsoon blowing determines the mixing grade of water columns at other stations. The northeast monsoon started in the autumn (October) and led to the occurrence of upper water column mixing (Figure 2A). This stirring effect was the most vigorous in the winter (January), and it resulted in the surface water temperatures reaching low values of 22 to 24 °C (Figure 2B). With the weakening of the monsoon, the water mixing gradually moderated in the spring (April). In the summer (July), significant stratification was observed and resulted in a high surface temperature greater than 29 °C (Figure 2C,D). Salinity was maintained above 34.5 at each station throughout the year (Figure 2A–D).

The nutrient distribution in the subtropical Kuroshio exhibited notably higher concentrations at the coastal upwelling stations. However, in the Kuroshio mainstream and oceanic province, the concentrations of both nutrients in the euphotic zones were extremely low (Figure 2E–L). In the autumn (i.e., October 2012) (Figure 2E) and the summer (i.e., July 2013) (Figure 2H), the depth of the nitracline gradually increased from the coast to the open sea. In contrast, in the winter (i.e., January 2013) (Figure 2F) and the spring (i.e., April 2013) (Figure 2G), except for the upwelling stations, the nitraclines were maintained at a depth of approximately 100 m. The seasonal distribution of PO_4_ was similar to that of NO_3_. The elevation of the nitracline derived from the mixing effect caused by the northeast monsoon in winter and spring would facilitate the upward transport of deep-sea nutrients. The chlorophyll was around 0.3 to 1.2 mg m^−3^ across four seasons. In the winter and spring, there was an even distribution of chlorophyll in the upper layer (Figure 2N,O). However, the maximum chlorophyll layer in the summer and autumn occurred at water depths of approximately 50 m and 100 m, except for that at the upwelling station (K1), which was found at the surface (1.29 mg Chl *a* m^−3^) (Figure 2M,P). Furthermore, the highest total content of Chl *a* (186.36 mg Chl *a* m^−3^) within the depth ≤ 100 m of most stations was found in the winter (Figure 2N).

The dominance of *Synechococcus* and picoeukaryotes was found in the surface water in both the cold and warm seasons (Figure 3A–D,I–L). The highest abundance of *Synechococcus* and picoeukaryotes both occurred at 10 m at the K1 station in the summer and spring, respectively (1.5 × 10^5^ cells mL^−1^ and 2.2 × 10^4^ cells mL^−1^) (Figure 3C,D,K,L). However, the maximum total cell numbers of *Synechococcus* and picoeukaryotes in the upper water column (≤100 m) appeared in the winter (9.3 × 10^8^ cells cm^−2^ and 2.3 × 10^8^ cells cm^−2^) (Figure 3F,J). In contrast to *Synechococcus* and picoeukaryotes, *Prochlorococcus* was found in all seasons (Figure 3E–H). Especially high total *Prochlorococcus* concentrations in the upper water column (≤100 m) were present in summer and autumn (3.3 × 10^9^ cells cm^−2^ and 2.4 × 10^9^ cells cm^−2^) (Figure 3E,H). In addition, it had the highest number at specific depths (75 m and 50 m) in these two seasons. In contrast, the distribution of *Prochlorococcus* was more even at depths ≤100 m in winter than in summer and autumn, although the total concentration of *Prochlorococcus* in the upper water column (≤100 m) was the same as that in autumn (2.9 × 10^9^ cells cm^−2^) (Figure 3F). According to the conversion factors of Chl *a* concentration of *Synechococcus* and *Prochlorococcus* (1.065 fg Chl *a* cell^−1^ and 1.51 fg Chl *a* cell^−1^) [52], their contributions to total Chl *a* in the upper water column (≤100 m) at different sampling sites are shown in Figure 4. On average, picocyanobacteria (*Synechococcus* and *Prochlorococcus*) were responsible for approximately 30% to 50% of the total Chl *a* in the four seasons; the highest percentage was shown in September 2009 and August 2015 (Figure 4C). *Synechococcus* contributed the lowest percentage of total Chl *a* in October (1.5%), while the highest contribution was in September (Figure 4A). *Prochlorococcus* contributed the highest percentage of total Chl *a* in the summer (August, 31.7%) (Figure 4B).

The total abundance of *Prochlorococcus, Synechococcus,* and picoeukaryotes at depths above 100 m had distinct correlations with biotic and abiotic environmental factors in the RDA (Figure 5). For example, *Prochlorococcus* was positively correlated with nitracline depth, while *Synechococcus* was negatively correlated with water temperature and the depth of the euphotic zone.

The environmental DNA was collected in K2 and K4 to represent the sample in the mainstream of the Kuroshio current when the K5 and K6 closed to the Kuroshio current. The K8 station was the oceanic station. A total of 4,410,567 high-quality reads were obtained after eliminating short- and low-quality sequences. The total reads had 1.6 to 9.8 × 10^5^ reads at each station. The overall coverage at each station was greater than 99%. Each station obtained 996 to 3012 ASVs (Table 1). The diversity and richness indices increased following the distance of the station away from the coast. Open ocean stations had higher diversity and richness indices of picoplankton compared with the stations affected by the Kuroshio current. Furthermore, the open ocean station (K8) had the highest richness and diversity indices, while the lowest indices were at the Kuroshio current station (K4) in the summer (Table 1). Hierarchical clustering analysis showed that the total bacterial community composition in the surface layer was different from that in the DCM layer (Figure 6). This result suggested that niche partitioning in the community occurred (ANOSIM, R = 0.68, *p* = 0.001).

In terms of the community structure of picoplankton at the surface of the Kuroshio current, the ASVs were categorized into the phyla *Cyanobacteria*, *Proteobacteria*, *Actinobacteria*, and *Bacteroidetes* (Figure 7). Each taxon occupied a consistent ratio in total reads among all stations during the four seasons. In detail, the reads affiliated with *Cyanobacteria* occupied 20 to 40% of the total reads. The highest cyanobacteria percentage occurred at the K4 station in each season. The maximum percentage of cyanobacteria at the K4 station was in the winter (41.6%) (Figure 7). *Proteobacteria* consistently occupied approximately 40% to 50% of the total reads (Figure 7). The major four genera of *Proteobacteria* were *Alphaproteobacteria*, *Deltaproteobacteria*, *Gammaproteobacteria*, and *Flavobacteria*. The most dominant genus was *Alphabacteria,* which accounted for approximately 20% to 30.8% of the total picoplankton during the four seasons. *Actinobacteria* accounted for only 6.6 to 14.6% of the total picoplankton in the four seasons.

The ASV reads for *Synechococcus* and *Prochlorococcus* in the total cyanobacteria phylum showed different seasonal patterns in both the surface and DCM layers (Figure 8). In detail, *Prochlorococcus* accounted for 71.1% to 92.2% of the total cyanobacteria in summer (Figure 8D,H) and autumn (Figure 8A,E) in the surface and DCM layers. In winter, the relative abundance of *Prochlorococcus* significantly decreased (Figure 8B,F) in the two layers. In particular, *Prochlorococcus* in DCM accounted for only 20% to 40% of the total cyanobacteria when it accounted for 30% to 80% on the surface (Figure 8). Then, the relative abundance of *Prochlorococcus* increased again in spring (40% to 60% of total cyanobacteria) in the two layers. In contrast, *Synechococcus* had a high percentage in winter and spring, accounting for approximately 20% to 80% of the total cyanobacteria, while that in summer and autumn accounted for only 2.3% to 13.8% in the two layers (Figure 8). Within the *Synechococcus* group, clades I, II, III, and VII formed the major group in the main Kuroshio current (Figure 9). *Synechococcus* clade II had the highest relative abundance of *Synechococcus* (90%) in four seasons of the surface (Figure 9A–D). *Synechococcus* clade II was also dominant in January and April of DCM (Figure 9F,G). However, it decreased by approximately 50% in the DCM warm seasons when the relative abundance of clade VII increased (Figure 9H). *Prochlorococcus* was categorized into two groups, the high light group (HL) and the low light group (LL), by the differential Chl *a* intensity in flow cytometry. The HL-II groups existed in both the surface and DCM and were responsible for more than 90% of the total relative abundance of *Prochlorococcus* (Figure 10). The HL-I group was mainly observed in the DCM and had a high relative abundance in the summer (Figure 10E–H). Finally, the LL group only presented DCM (Figure 10E–H). Interestingly, the relative abundance of the LL group started to increase in spring and had a maximum relative abundance in the summer (approximately 20%) (Figure 10G,H).

## 4. Discussion

The Kuroshio current is an important western boundary current in the Northwest Pacific Ocean. More than half of Chl *a* in the Kuroshio current is contributed by picoplankton [29]. However, in comparison with other highly oligotrophic oceanic currents (i.e., the Gulf of Mexico), few studies have focused on prokaryotic picoplankton in the Kuroshio current [54,55,56]. Thus, in this study, we revealed the detailed community structure of picoplankton and their distribution in the Kuroshio current across four seasons. It would help to understand the correlation between the western boundary current hydrography and the picophytoplankton succession.

The Kuroshio current experiences sudden nutrient input events that boost *Synechococcus* abundance. In our in situ observation results, the highest abundance of *Synechococcus* primarily appeared at the K1 station in the summer (1.5 × 10^5^ cells mL^−1^) and the spring (9.9 × 10^4^ cells mL^−1^), where there was strong nutrient input from upwelling (coastal uplift). On the other hand, obvious stratification was found at other stations. Thus, the nutrients at the other stations were scarcer than those at the K1 station and could not support *Synechococcus*, which had a high abundance similar to that at the K1 station. Liu et al. (2021) indicated that the growth of *Synechococcus* in the Kuroshio current was enhanced following increasing temperature by dilution experiments [57]. In addition, under high temperature (surface water temperature + 4 °C), a higher growth rate of *Synechococcus* was observed in nutrient-replete conditions than in nutrient-limited water [57]. Hence, in the summer of the Kuroshio current, sudden nutrient input events could induce *Synechococcus* to thrive temporarily. It has also been demonstrated that the nutrients brought by dust storms, typhoons, and coastal uplift promote the growth of *Synechococcus* [23,24,29]. In addition, deeper nutrients uplifted by upwelling also stimulated *Synechococcus* to grow in the upwelling area [58,59].

Another finding was that the total cell number of *Synechococcus* in the upper water column (≤100 m) was higher in the winter than in the summer. From a previous study, the growth rates of *Synechococcus* increased with increasing water temperature. Although the average surface water temperature in summer was 28.7 °C, it remained at 26.1 °C in the winter season of the Kuroshio current. Following a previous study, the growth rate of *Synechococcus* at 26 °C remained comparable to that at 28 °C. Thus, we suspected that the growth rate of *Synechococcus* remained in these two seasons. Furthermore, because of monsoon-induced vertical mixing, the nutrients in the winter were transported more effectively to the surface layer in the winter. Additionally, there was strong upwelling invasion during cold seasons. The upwelling events were observed not only at coastal stations (K1 and K2) but also at open ocean stations (K8 and K9). From the above, these factors likely contributed to a more evenly distributed number of *Synechococcus* across each station in the winter. In fact, the highest number of *Synechococcus* in winter reached 9.3 × 10^4^ cells mL^−1^. Consequently, the total cell number of *Synechococcus* in the upper water column (≤100 m) was the highest during the winter season (9.3 × 10^8^ cells cm^−2^). This explains the negative correlation between water temperature and the abundance of *Synechococcus* in the RDA. Monsoon-induced vertical mixing, upwelling, and holding high water temperature in the Kuroshio current caused even distribution and high total *Synechococcus* abundance in the upper water column (≤100 m) in the winter.

Offshore of northeastern Taiwan, upwelling often occurs when the branch of the Kuroshio current intrudes into the East China Sea shelf at higher latitudes. Chung and Gong (2019) [58] discovered that the surface of upwelling sites exhibited a high abundance of *Synechococcus* (5.9 × 10^4^ cells mL^−1^), ranging from 1 to 2 × 10^4^ cells mL^−1^ in the surface waters of the sites influenced by the Kuroshio current. The relative abundance of *Synechococcus*, based on total 16S rRNA amplicon sequencing, was up to 96%. These *Synechococcus* populations contained highly phylogenetic divergence, including clade II, clade X, and clade XI [58], with clade II being the most dominant (96% of total *Synechococcus*). In this study, we focused on a more southern Kuroshio current, characterized by low available nutrients in situ. Therefore, the differences found in the abundance of *Synechococcus* compared with the findings of Chung and Gong (2019) [58] could be attributed to variations in the formation of upwelling caused by the main or branch currents of the Kuroshio current system [58]. Additionally, this study revealed the presence of *Synechococcus* in clade VII, clade III, and clade X. These variations in clade composition are likely influenced by different depths of upwelling, which might provide distinct nutrient concentrations for microorganisms in situ [60,61,62].

Picoeukaryotes may also be stimulated by nutrient input. During the winter, picoeukaryotes and *Synechococcus* and picoeukaryotes both exhibited high abundance, attributed to nutrient input from subsurface layers through mixing. In addition, picoeukaryotes were also particularly abundant on the surface of upwelling stations (K1) during the spring and summer. The RDA revealed a positive correlation between the abundance of picoeukaryotes and *Synechococcus* (Figure 5), supporting the notion that the two microorganisms are correlated. Chan et al. (2020) also demonstrated that the abundance of picoeukaryotes in the Kuroshio current was positively correlated with *Synechococcus* abundance (Pearson correlation, *p* < 0.01) [63].

*Prochlorococcus* is dominant in many oligotrophic environments, such as central oceanic gyres and the southern Gulf of Mexico [64,65,66]. It has also consistently been observed as the prevailing group upstream of the Kuroshio current [57,67]. According to our results, the total abundance of *Prochlorococcus* in the upper water column (≤100 m) showed a positive correlation with the nitracline depth (*p* = 0.031). This indicates that the total *Prochlorococcus* increased as the nitracline depth increased, which has a scarce nitrate concentration. This relationship was particularly evident during summer and autumn, characterized by a deeper nitracline compared to other seasons (summer: R^2^ = 0.80 and *p* < 0.05; autumn: R^2^ = 0.75 and *p* < 0.05). Notably, while *Prochlorococcus* exhibited a more widespread vertical distribution in other seasons, its distribution was concentrated at specific depths within the DCM during summer and autumn. Furthermore, the relative abundance of the LL and HL-I *Prochlorococcus* ecotypes, which thrive in low temperature and low light intensity, increased in DCM during summer and autumn (Figure 10H,E) [68]. In contrast, the LL and HL-I ecotypes displayed similar relative abundances in the winter season, possibly due to well mixing that brought them to the surface (Figure 10B,F). Recent studies have revealed that *Prochlorococcus* carries numerous nitrate assimilation genes and is abundant in or near nitracline in oligotrophic marine environments [55,69,70]. Therefore, this distinct difference in abundance is likely attributable to a clear stratification in the subsurface and deep nitracline during summer and autumn that allows the two *Prochlorococcus* ecotypes at deeper depths to rapidly utilize nutrient pulses [71].

## 5. Conclusions

The nutrient-scarce, warm, and high-salinity Kuroshio current has a profound impact on both the marine ecology of the northwestern Pacific Ocean and the global climate [72,73,74]. It is important to understand the characteristics of the fundamental microorganism community in different regions within the Kuroshio current. This study revealed that the composition of prokaryotic picoplankton was significantly different between the surface and DCM, except in January, which had a deep mixing zone. *Synechococcus* (dominated in Clade II) and *Prochlorococcus* (dominated in HL-II groups) were the major members of picocyanobacteria, which accounted for half of the Chl *a* in the Kuroshio current. The seasonal dynamics of *Synechococcus* were caused by water temperature, nutrient input, and euphotic zone, whereas *Prochlorococcus* had a positive correlation with nitracline depth. Thus, nutrients in situ rapidly and highly affected the seasonal dynamics of these fundamental microorganism groups. Under the scenario of global warming, the flow-speed of Western Boundary Currents would be decelerating [74]. It would affect the nutrient transport of Western Boundary Currents and in turn change the abundance and distribution of picoplankton in these regions. The long-term observation of the variations in picocyanobacterial assemblage and their relative biotic and abiotic factors can highlight their importance in the ocean, which is regulated by global warming [9].

## Figures and Tables

**Figure 1 biology-12-01424-f001:**
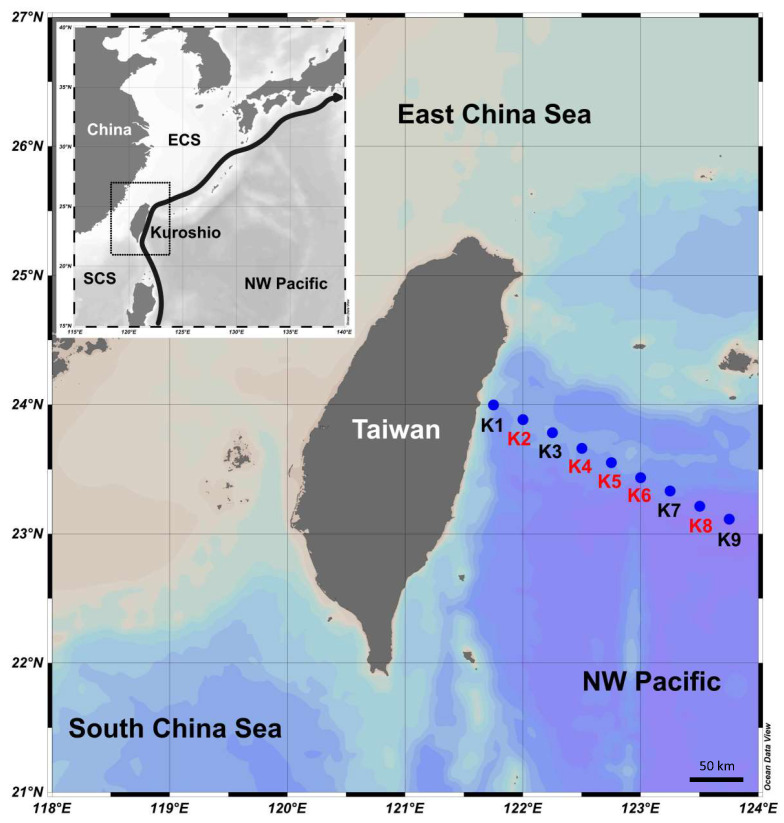
Sample collection sites (K1–K9) in eastern Taiwan. The words in red indicate the sampling stations with environmental DNA data.

**Figure 2 biology-12-01424-f002:**
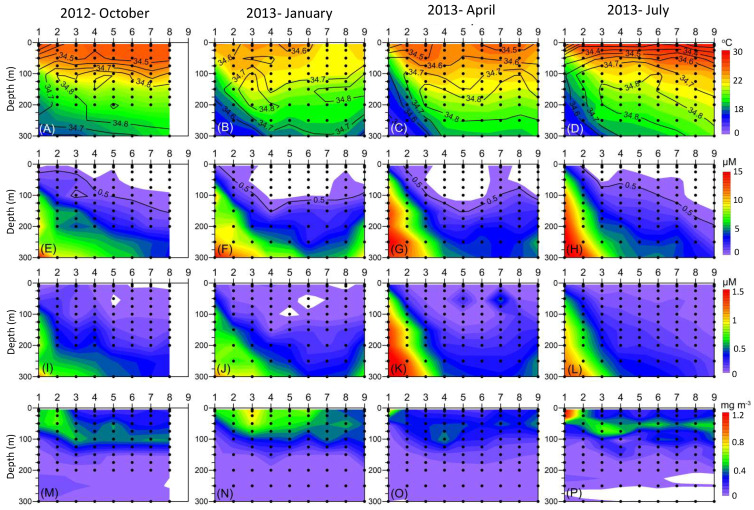
Vertical profiles of water temperature (°C, **A**–**D**), nitrate (μM, **E**–**H**), phosphate (μM, **I**–**L**), and Chl *a* concentration (mg m^−3^, **M**–**P**) along Stations K1 to K9 during October 2012 and January, April, and July 2013. The black lines in (**A**–**D**) are salinity. The nitrate clines (black line) are indicated in (**F**–**I**).

**Figure 3 biology-12-01424-f003:**
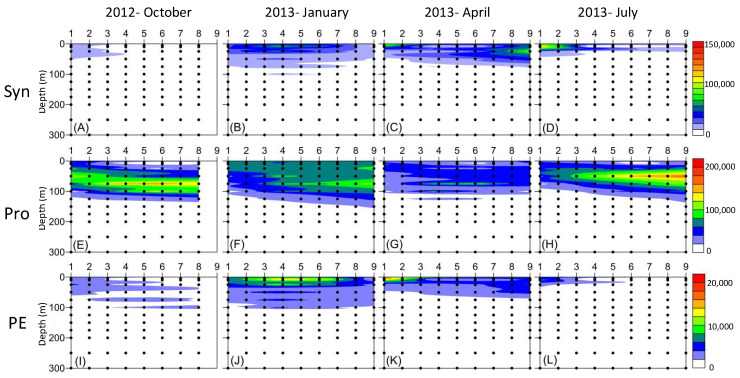
Vertical profiles of the abundance of *Synechococcus* (Syn, **A**–**D**), *Prochlorococcus* (Pro, **E**–**H**), and picoeukaryotes (PE, **I**–**L**) along Stations K1 to K9 during October 2012 and January, April, and July 2013.

**Figure 4 biology-12-01424-f004:**
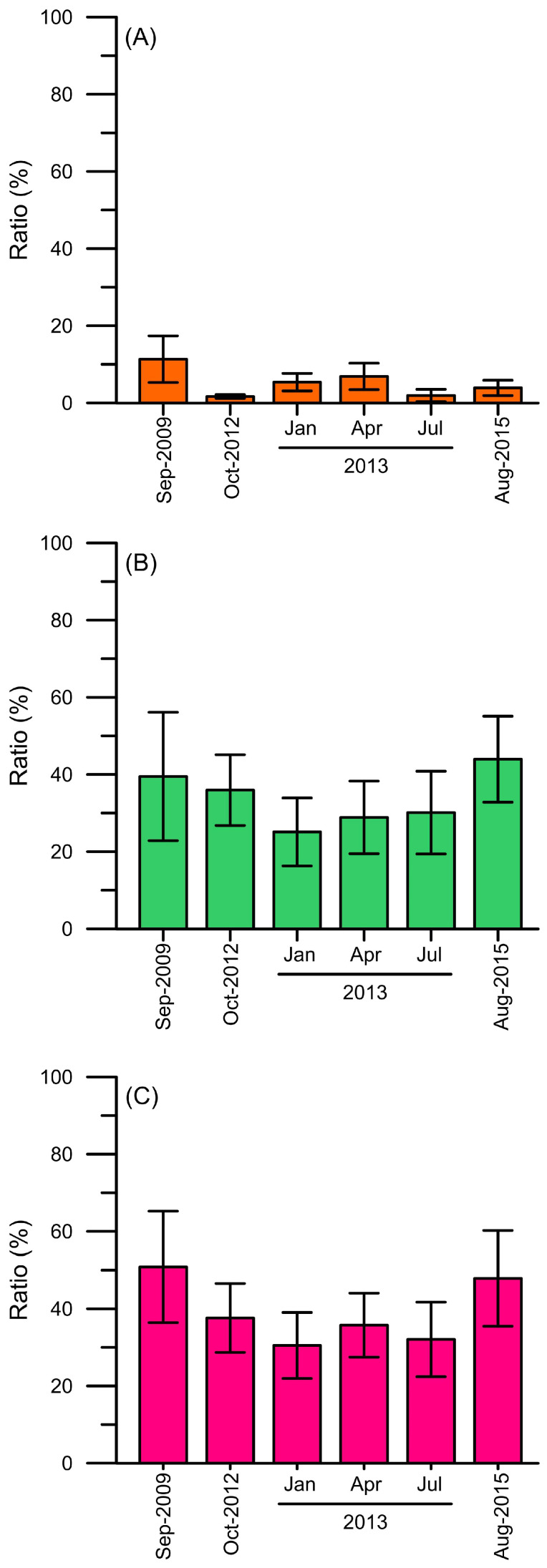
Chl *a* content in the (**A**) *Synechococcus,* (**B**) *Prochlorococcus*, and (**C**) picocyanobacteria (*Synechococcus* and *Prochlorococcus*) of total Chl *a* (%) in the upper water column (≤100 m) from September 2009, October 2012, January, April, July 2013, and August 2015. Standard deviation is shown.

**Figure 5 biology-12-01424-f005:**
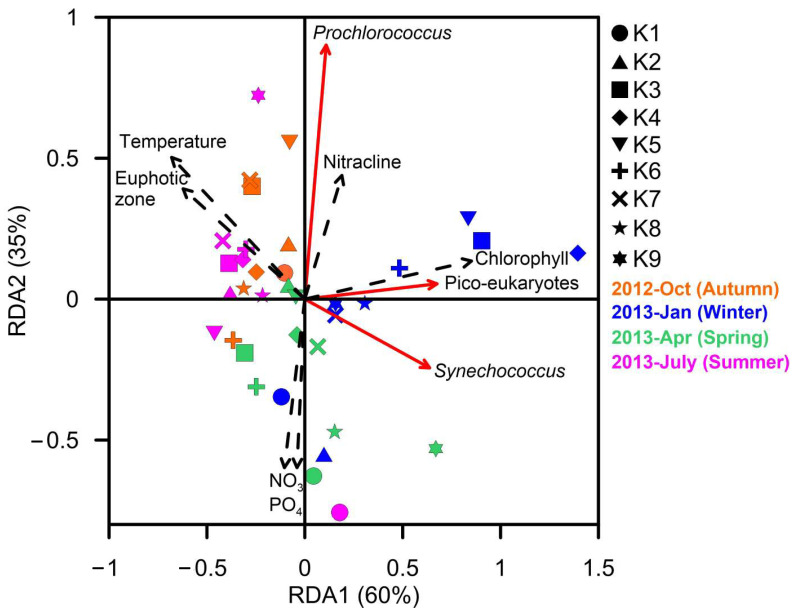
Redundancy analysis (RDA) of the relationship of environmental factors (black dashed line arrows) and total abundance of different picocyanobacteria groups (red line arrows) in the upper water column (≤100 m) of different stations during the four sampling months.

**Figure 6 biology-12-01424-f006:**
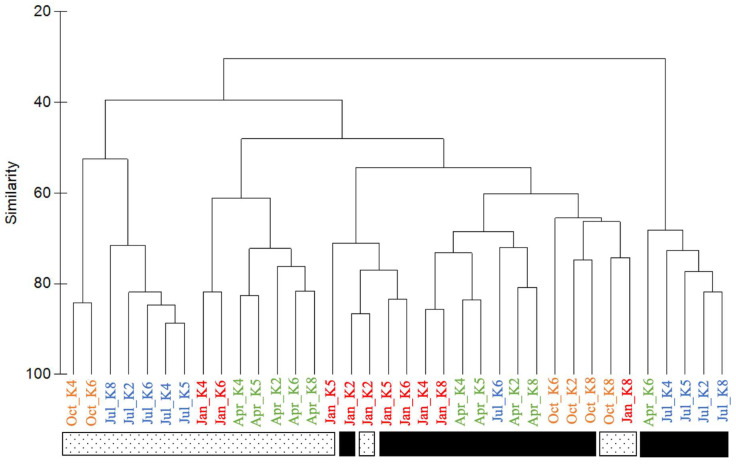
Dendrogram from hierarchical cluster analysis for all ASVs from surface (dots rectangle) and DCM (black rectangle) in each station of different months (words in color). The Bray–Curtis similarity index is used for clustering.

**Figure 7 biology-12-01424-f007:**
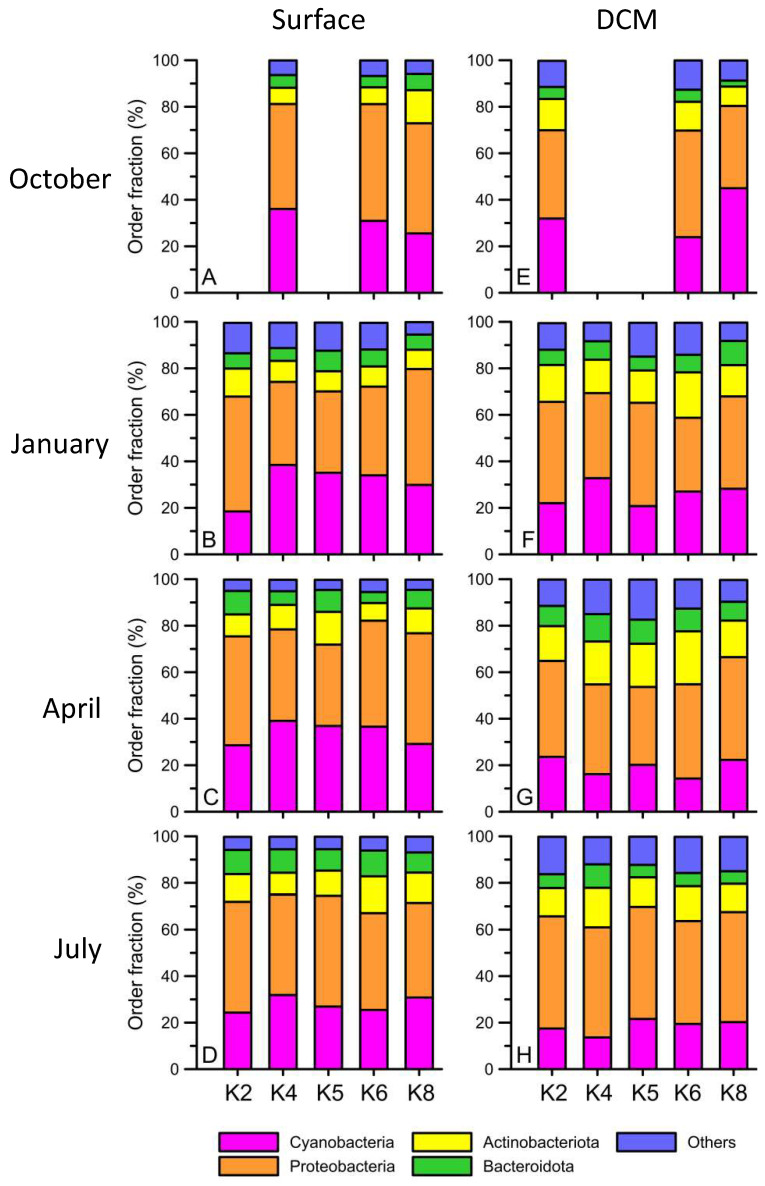
The hierarchical order fraction (%) of different bacterial communities (in order level) from the surface (**A**–**D**) and DCM (**E**–**H**) at Stations K2, K4, K5, K6, and K8 during October 2012 (**A**,**E**), January (**B**,**F**), April (**C**,**G**), and July 2013 (**D**,**H**).

**Figure 8 biology-12-01424-f008:**
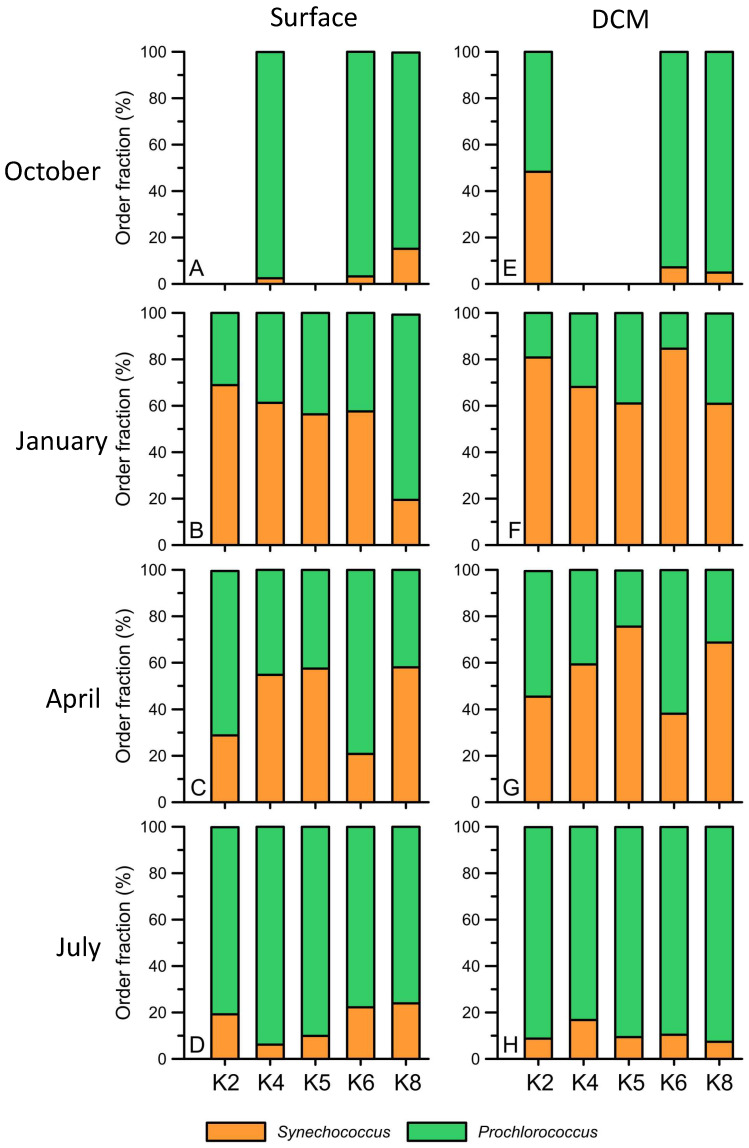
The hierarchical order fraction (%) of *Synechococcus* and *Prochlorococcus* from the surface (**A**–**D**) and DCM (**E**–**H**) at Stations K2, K4, K5, K6, and K8 during October 2012 (**A**,**E**), January (**B**,**F**), April (**C**,**G**), and July 2013 (**D**,**H**).

**Figure 9 biology-12-01424-f009:**
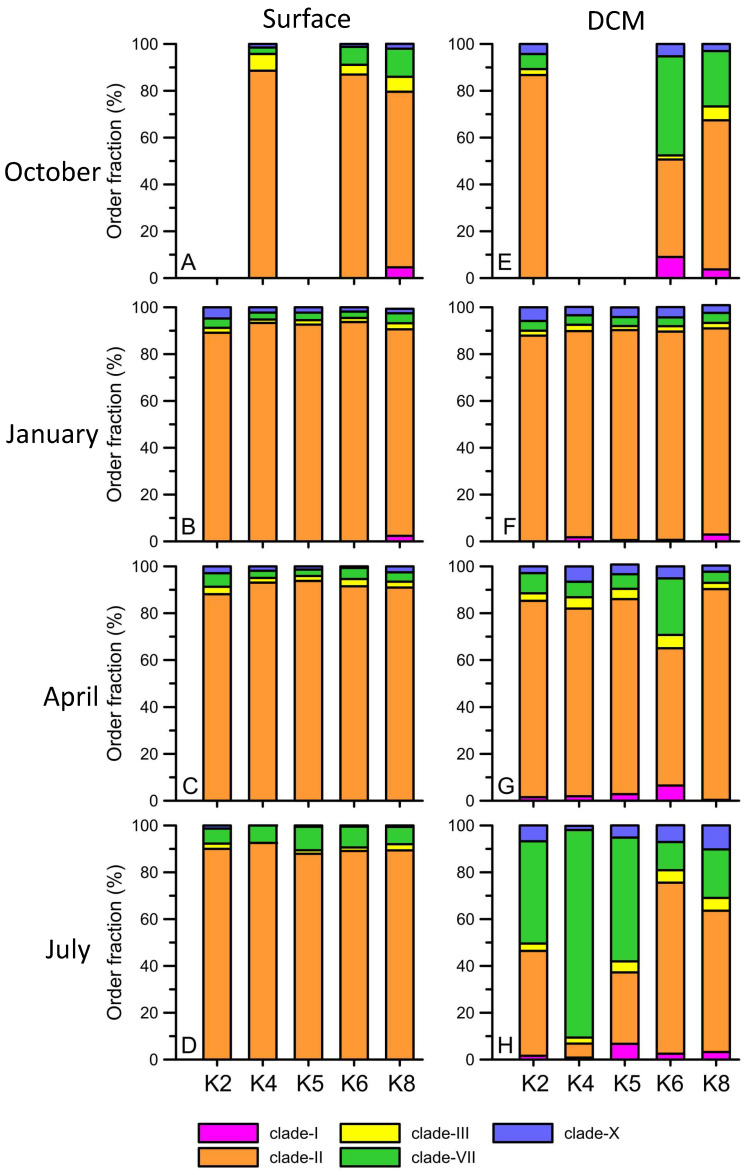
The hierarchical order fraction (%) of *Synechococcus* from the surface (**A**–**D**) and DCM (**E**–**H**) at Stations K2, K4, K5, K6, and K8 during October 2012 (**A**,**E**), January (**B**,**F**), April (**C**,**G**), and July 2013 (**D**,**H**). Different colors indicate different clades (clade-I, clade-II, clade-III, clade-VII, and clade-X).

**Figure 10 biology-12-01424-f010:**
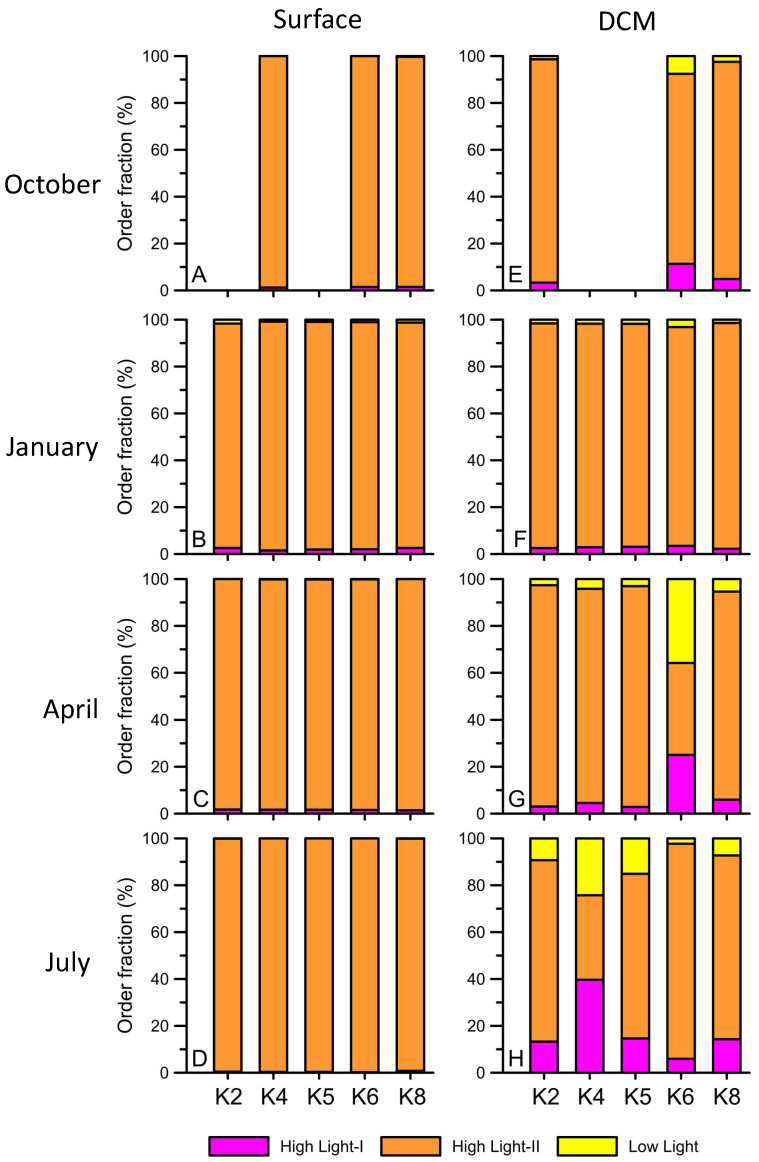
The hierarchical order fraction (%) of *Prochlorococcus* from the surface (**A**–**D**) and DCM (**E**–**H**) at Stations K2, K4, K5, K6, and K8 during October 2012 (**A**,**E**), January (**B**,**F**), April (**C**,**G**), and July 2013 (**D**,**H**). The three different colors indicate high light-I, high light-II, and low light groups.

**Table 1 biology-12-01424-t001:** The results of 16S rRNA gene V3-V4 sequencing and the indices of species richness (ACE) and diversity (Shannon).

Time	Station	Reads	ASV	ACE	Shannon
October 2021	K4	161,662	996	997	4.74
	K6	199,278	1162	1162	4.95
	K8	319,142	1551	1552	5.11
January 2013	K2	427,192	2082	2083	5.71
	K4	407,216	1807	1808	5.01
	K5	976,914	3012	3014	5.38
	K6	272,391	1590	1590	5.22
	K8	367,891	1764	1765	5.36
April 2013	K2	291,525	1335	1335	5.07
	K4	206,446	1233	1233	5.16
	K5	312,912	1409	1409	4.73
	K6	294,990	1235	1235	4.65
	K8	222,253	1201	1201	4.88
July 2013	K2	251,552	1235	1236	4.97
	K4	283,202	1139	1139	4.75
	K5	289,476	1134	1134	1.88
	K6	278,805	1161	1162	4.85
	K8	626,538	1898	1899	5.03

## Data Availability

The nucleotide sequences used in this study have been deposited in the Sequence Read Archive (SRA) database under BioProject accession number PRJNA904529.

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
