# Peer review of "Seasonal Patterns of Picocyanobacterial Community Structure in the Kuroshio Current"

_biology, 2023, doi:10.3390/biology12111424_

Round 1

Reviewer 1 Report

Comments and Suggestions for Authors

Evaluation of the ms. titled “Seasonal patterns and diversity of prokaryotic picoplankton community structure in the Kuroshio Current”, by Chan et al., submitted to be published in the international journal BIOLOGY.

The manuscript (ms.) evaluated deals with studies on the prokaryotic picoplankton in a very important current system during four different seasons. This ms. is basically descriptive, which in many occasions is necessary, as there are few or no previous backgrounds. However, despite that the title refers to the “community structure”, basically two main members of this community were studied, the cyanobacteria Synechococcus and Prochlorococcus. Other taxonomic groups were also studied (including picoeukaryotes), but they didn’t take an important part in the results and especially discussion. Additionally, the seasonal patterns were described only for one year, and they are probably not representative. Therefore, it is a little bit disappointing to find that the title does not reflect the content of the ms. Finally, the text as it is presented led to confusion as some terms are used indistinctly and there are several omissions and mistakes along the ms.

There are other small and big issues along the ms. which are annotated as follows:

- Last paragraph of the Abstract is not supported by the information provided, and it should be rewritten or frankly deleted.

- In the Introduction only picoplanktonic photosynthetic cyanobacteria are mentioned. All other non-photosynthetic groups (diverse bacteria) were not included, despite the current work also deals with Proteobacteria and Actinobacteria.

- Material and methods: Depths of biological sampling are NOT clearly established.

- Results: More precision is required in indicating the exact figure where authors are describing (e.g. the maximum chlorophyll later, Fig. 2 M-P ?).

Not clear what are the other groups (“prokaryotic picoplankton”) contributing to the total chlorophyll a (Fig. 4). They were not previously mentioned; I suppose it is a mistake and it should be regarded as “picoeukaryotes”.

L. 276- “bacterial community” Is this part of the picoplankton studied ? Not clear.

How the different clades and groups of both Synechococcus and Prochlorococcus were determined ?

- Discussion:

L. 433 and on- DNA extraction methods as a reason for finding “different compositions of the picoplankton” ? This is a serious subject that must be tested.

Most of the last paragraph should be rewritten or deleted, as it is very speculative.

I must say that the format the manuscript is presented does not permit clearly differentiate parts of the text without interference of figures or tables.

I am very sorry, but considering all mentioned above, I do not recommend publication of the ms. in its present version. The points raised above should be considered to produce a new, improved version.

Comments on the Quality of English Language

No major problems with that.

Author Response

Dear Editor,

I am submitting a revised manuscript entitled “Seasonal Patterns of Picocyanobacterial Community Structure in the Kuroshio Current” by Ya-Fan Chan, Chih-Ching Chung, Gwo-Ching Gong, I-Jung Lin, and Ching-Wei Hsu for consideration as a publication in Biology. In this version, the title of this manuscript has been modified. In addition, we have carefully revised this manuscript one by one in accordance with the reviewers' comments. We sincerely hope that you and the 3 reviewers will be satisfied with the revised version.

Sincerely,

Chih-Ching Chung

Associate Professor

Institute of Marine Environment and Ecology

National Taiwan Ocean University

Keelung, 20224, TAIWAN

TEL: +886-2-24622192 ext. 5704

Reviewer 2 Report

Comments and Suggestions for Authors

Please see section-by-section and point-by-point comments and suggestions here, along with the annotated PDF.

Revision for Ms. Biology [MDPI] # 2661811

Title: The current title is “Seasonal Patterns and Diversity of Prokaryotic Picoplankton Community Structure in the Kuroshio Current”. In Ecology, the term diversity may refer to many things (from species richness, which is a simple measure of diversity to γ-diversity at the landscape scale; on the other hand, community structure may be described from several standpoints as well. Furthermore, diversity and community structure need not necessarily be connected; thus I suggest the title be reformulated (may I suggest the very simple addition of the term “and” linking prokaryotic picoplankton diversity and community structure?)

Abstract: the abstract clearly describes the core of the manuscript. It lacks a clear statement of the aims of this research

Keywords: 2 out of the 4 keywords are listed on the title; that is a bit redundant

Introduction: comments were done in the form of highlighting and call-out notes on the original PDF. In general, it is very well structured, providing background information of picophytoplankton on a global scale, their projected trends in climate change scenarios; and then narrowing the discussion to Synechococcus and Prochlorococcus, to end up with an introduction to the Kuroshio system in the last paragraph.

A few other (specific and quite important) comments follow:

First paragraph to line 46: provides a good background for your research. This opening paragraph should be closed with a clear statement of the aims of your study. You may also consider phrasing it as a hypothesis, so that the whole research doesn’t sound like a mere sampling and monitoring. The same applies to the closing of the Introduction section (lines 93-96)

Materials and Methods: the methods (in the field, lab, data processing and statistical analyses) are accurately described and appropriate for this study.

Results:

Line 204: this information seems redundant; as vertical mixing wouldn’t affect salinity

Lines 211-14. Unclear; re-phrase the sentence

Line 221: the style is too coarse; re-write the sentence. Also, this is more of an interpretation, rather than a result per se… It should be in the Discussion.

Discussion:

In general, the discussion is well structured, going from the general to the specifics.

Lines 346-48 not clear.

Lines 436-445: raises an interesting point for discussion. Have the authors considered (or carried out) other methods for estimating floristic composition and density in picoplakton, e.g., cell counts with a flow cytometer?

Figures:

Fig. 1: latitude and longitude are ok, but add scale in km

Fig. 2: this is an excellent composite of data. The figure is very informative and easy to follow. Please add scale in y-axis (depth in m)

Fig. 3: ditto for Fig. 2

Comments on the Quality of English Language

I detected some awkward phrasing here and there, but pointed it out.

Reviewer 3 Report

Comments and Suggestions for Authors

The result is certainly within the expectation based on literature I read. But I really want to ask, it has been about 10 years since these old data were obtained, now is it not a good timing to repeat the measurements and make comparisons? I don't think this paper is interesting to the readers, because it doesn't present anything related to climate change you emphasized in the introduction. I sincerely suggest holding the data for a new paper. 

Minor comment:  Show us the unit in the caption or the legend of Figure 2, and it's really hard to see the numbers of the legend.

Round 2

Reviewer 3 Report

Comments and Suggestions for Authors

My point is clear about this manuscript's weakness, it's a data report. The authors also clearly stated their difficulties in obtaining more data. I will not oppose if editors decide to publish it.